# You Only Sample (Almost) Once: Linear Cost Self-Attention Via Bernoulli Sampling

## Abstract

Transformer-based models have come to dominate the landscape in a wide range of natural language processing (NLP) applications. The heart of the transformer model is the self-attention mechanism, which captures the interactions of token pairs in the input sequences and consequently, depends quadratically on the input sequence length. It is known that training such models on longer sequences is quite expensive, and often, prohibitively so. We show that a Bernoulli sampling attention mechanism based on Locality Sensitive Hashing (LSH), decreases the quadratic complexity to linear. We bypass the quadratic cost by considering self-attention as a sum of individual tokens associated with Bernoulli random variables that can, in principle, be sampled at once by a single hash (although in practice, this number may be a small constant). This leads to an efficient sampling scheme to estimate self-attention which relies on specific modifications of LSH (based on feasibility of deployment on GPU architectures). We evaluate our proposed algorithm on the GLUE benchmark with standard 512 sequence length and our method achieves comparable or even slightly better performance than a standard pretrained Transformer. To evaluate whether our method can indeed handle longer sequences, we conduct experiments on long sequence (4096) language model pre-training and achieve consistent results as standard self-attention, while observing sizable inference speed-ups and memory savings.

## 1 Introduction

The Transformer model (Vaswani et al., 2017) is incredibly effective across a diverse set of natural language processing (NLP) applications including machine translation (Vaswani et al., 2017), language inference (Devlin et al., 2018) and paraphrasing (Raffel et al., 2019). Transformer-based models such as BERT (Devlin et al., 2018) are pretrained in an unsupervised manner and later finetuned on different downstream tasks, often providing state-of-the-art performance on standard benchmarks. While such models have strong empirical performance, their high computational and memory requirements remain quite high. Consequently, in the NLP setting, most current models have certain constraints on the sequence length, e.g., BERT and other transformer-based language models (Yang et al., 2019; Liu et al., 2019) limit the sentence length to be at most $512$.

The Multi-Head Self-Attention is central to Transformer based models and provides a flexible global receptive field to exchange information among input tokens. While self-attention provides immense benefits, it is also a key bottleneck in training with long sequences. In particular, the output of self-attention is a combination of all tokens where coefficients are determined by the similarities among tokens. While this is empirically beneficial, it involves a sizable resource footprint. For sequence length $n$, this leads to a $O(n^2)$ complexity in both time and memory to compute pairwise similarities among all input tokens. This quadratic cost is a roadblock in attaining potential benefits that may be realizable in various applications by capturing long term context dependencies. As we will discuss in more detail later, the foregoing issue is a major thrust of several recent and ongoing efforts focused on mitigating the sizable resource requirements of such models.

Our work is inspired by ideas of importance sampling via hashing-based sampling techniques (Spring & Shrivastava, 2017; Charikar & Siminelakis, 2017). We proposed a Bernoulli based sampling to approximate self-attention, scaling linearly with the input sequence length. We achieve this by viewing self-attention as a sum of individual tokens associated with Bernoulli random variables

whose success probability is determined by the similarities among tokens. In principle, we can sample all Bernoulli random variables at once with a single hash (although in practice, this number may be a small constant to lower the approximation variance). This leads to an efficient sampling scheme to estimate self-attention which relies on specific modifications of hashing-based importance sampling (based on feasibility of deployment on GPU architectures). The resulting strategy (You Only Sample Almost Once, YOSO-Attention) is far more amenable to an efficient and backpropagation friendly implementation, and has a favorable empirical performance profile on natural language modeling tasks. We evaluate our proposed algorithm on the GLUE benchmark (Wang et al., 2019) with $512$ sequence length as well as on long sequence language model pretraining where we see promising results with speed-ups and memory savings.

## 2 BACKGROUND: SELF-ATTENTION

**Self-Attention.** Self-attention is a scaled dot-product attention mechanism to capture token dependencies in the input sequence, which can be defined as,

$$\mathcal{A}(\boldsymbol{Q}, \boldsymbol{K}, \boldsymbol{V}) = \text{softmax}\left(\underbrace{\frac{(\boldsymbol{Q}\boldsymbol{W}_Q)(\boldsymbol{K}\boldsymbol{W}_K)^T}{\sqrt{d_h}}}_{\mathcal{P}}\right)\boldsymbol{V}\boldsymbol{W}_V = \boldsymbol{D}_{\mathcal{P}}\exp\left(\mathcal{P}\right)\boldsymbol{V}\boldsymbol{W}_V \qquad (1)$$

where $\boldsymbol{Q}, \boldsymbol{K}, \boldsymbol{V} \in \mathbb{R}^{n \times d}$ are embedding matrices from the input sequence, called queries, key and values respectively. Here, $n$ is the input sequence length, $d$ is the embedding dimension of each token, $\boldsymbol{W}_Q, \boldsymbol{W}_K, \boldsymbol{W}_V \in \mathbb{R}^{d \times d_h}$ are learned parameter matrices, $d_h$ is the dimension of hidden embedding, and $\boldsymbol{D}_{\mathcal{P}}$ is a $n \times n$ diagonal matrix which normalizes each row of the $\exp\left(\mathcal{P}\right)$ matrix such that the row entries sum up to 1. For simplicity, we overload the notations for $\boldsymbol{Q}, \boldsymbol{K}, \boldsymbol{V}$ to denote $\boldsymbol{Q}\boldsymbol{W}_Q, \boldsymbol{K}\boldsymbol{W}_K, \boldsymbol{V}\boldsymbol{W}_V$ in our presentation.

**Multi-Head Self-Attention.** Multi-Head self-attention in Transformers runs through the scaled dot-product attention multiple times and the attention outputs are concatenated to help the model capture information from multiple representation subspaces Vaswani et al. (2017). Multi-Head Self-attention can be formally written as,

$$\text{MultiHead}(\boldsymbol{Q}, \boldsymbol{K}, \boldsymbol{V}) = \text{Concat}\big(\mathcal{A}_1(\boldsymbol{Q}, \boldsymbol{K}, \boldsymbol{V}), \cdots, \mathcal{A}_h(\boldsymbol{Q}, \boldsymbol{K}, \boldsymbol{V})\big)\boldsymbol{W} \qquad (2)$$

where $h$ is the number of heads, $\mathcal{A}_i, i = 1, \ldots, h$ are heads with different parameter matrices.

**Self-Attention Bottleneck.** A key bottleneck in self-attention is computing the softmax matrix, softmax$(\mathcal{P})$, which requires the calculation of all pairwise input token similarities. To reduce this cost, we seek to approximate the softmax matrix by viewing self-attention for each query as an expectation of a softmax distribution and computing the approximated self-attention with an efficient sampling mechanism. In the following sections, we will first review LSH-based importance sampling and then propose Bernoulli sampling with LSH to estimate self-attention efficiently.

## 3 IMPORTANCE SAMPLING VIA LOCALITY SENSITIVE HASHING

Importance sampling (Press et al., 2007) helps approximate properties of a target distribution by a weighted average of random draws from another distribution. It is known (Press et al., 2007) that importance sampling can be directly used for the softmax distribution by drawing samples from a uniform distribution – which avoids sampling from the softmax distribution directly which is harder. But this leads to a high variance estimate since the softmax distribution is usually concentrated in a small region. When using this idea for softmax matrix approximation for *self-attention in particular*, the variance tends to grow with the input sequence length. Before proceeding, we will summarize an interesting importance sampling method for low variance estimators, specifically, importance sampling via LSH from (Charikar & Siminelakis, 2017; Spring & Shrivastava, 2017).

**LSH-based Importance Sampling**. Consider the case when the angular distance between a key and a query is small. In this case, the similarity (between the key and the query) as well as the softmax probability will be large. When viewed through the lens of a nearest neighbor retrieval, the above property coincides with a large collision probability of high similarity key-query pairs, assuming that

the neighbor retrieval is implemented via LSH. Motivated by the link between softmax probability $p$ and LSH collision probability $q$, Spring & Shrivastava (2017) and Charikar & Siminelakis (2017) suggest using LSH as an efficient sampler for low variance softmax estimators.

**(a)** Spring & Shrivastava (2017) propose approximating softmax by sampling a set, $S$, a collection of neighboring keys for each query formed by the union of colliding keys using $m$ hash tables. The estimator is computed using $|S|^{-1} \sum_{i \in S} \frac{p(\boldsymbol{q}, \boldsymbol{k}_i)}{q(\boldsymbol{q}, \boldsymbol{k}_i)} \boldsymbol{v}_i$ where $\boldsymbol{q}$ is a query vector, $\boldsymbol{k}_i, \boldsymbol{v}_i$ are key and value vectors in the sampling set $S$, and $p(\cdot, \cdot)$ and $q(\cdot, \cdot)$ are softmax probability and collision probability of given pairs. The procedure is equivalent to performing importance sampling without replacement, which involves a dependency among the samples. Deduplication (avoiding double counting) requires memory to store keys in each hash table and runtime to deduplicate keys for each query. If the size of hash buckets is skewed, the GPU memory needs depend on the size of the hash bucket and the runtime depends on the size of $S$.

**(b)** Charikar & Siminelakis (2017) proposed a Hash based Estimator to simulate a proposal distribution for importance sampling via LSH, which can be easily applied in the context of softmax. For each hash table, a key is uniformly selected from the bucket that the query is hashed to, for simulating a draw from a proposal distribution. The estimate is computed as $m^{-1} \sum_{i=1}^{m} \frac{p(\boldsymbol{q}, \boldsymbol{k}_i)|H_i(\boldsymbol{q})|}{q(\boldsymbol{q}, \boldsymbol{k}_i)} \boldsymbol{v}_i$ where $|H_i(\boldsymbol{q})|$ denotes the size of hash bucket in the $i$-th hash table which $\boldsymbol{q}$ is hashed to. This simulates $m$ samples drawn with replacement from the proposal distribution. However, the probability of one key being sampled depends not only on (a) the angular distance to the query but also (b) the number of keys within the hash bucket, leading to a sampling dependency among all keys. Further, using it for self-attention causes a dependence between the sparsity in the softmax matrix and the number of hashes used. Specifically, the number of tokens that each query can attend to is bounded by the number of hashes: the procedure samples at most one key for each hash table and so, it adds one additional nonzero to the softmax matrix, at most.

*Remark 1.* While LSH-based importance sampling exploits the agreement between high probability $p(\cdot, \cdot)$ and high collision probability $q(\cdot, \cdot)$, the alignment is not perfect. Samples from proposal distribution must be reweighted to compensate for the difference. Further, for different queries, the likelihood ratios between softmax distribution and proposal distribution w.r.t. a single key are different. Therefore, the reweighing has to be done *during* querying. Although maintaining hash tables for storing keys is not a major problem in general, the high memory cost for hash tables and computation time for reweighing would influence efficiency when applied to self-attention.

## 4 YOSO-ATTENTION

We start from LSH-based importance sampling and seek to address some of the aforementioned issues when it is deployed for approximating self-attention. Instead of using LSH to simulate sampling from a proposal distribution over tokens, we view attention as a *sum* of tokens associated with Bernoulli random variables. This modification relates better to LSH and less with LSH-based importance sampling – the probability of one query colliding with a key is not based on other keys. This strategy helps avoid the sampling dependency issue in LSH-based importance sampling and offers an opportunity to develop a strategy more amenable to GPUs.

*Remark 2.* We assume that the input keys and queries of self-attention are unit length – to unify dot-product similarity in self-attention and cosine similarity in LSH. This is simple using Neyshabur & Srebro (2015): a temperature variable $\tau$ is used to bound the squared $\ell_2$ norm of all queries and keys and to reconstruct new unit length keys and queries while preserving their pairwise similarities. We can work with the softmax matrix in angular distance metric and derive our algorithm.

**Self-Attention via Bernoulli Sampling**. We aim to approximate self-attention, which uses a softmax matrix to capture the context dependency among tokens via their pairwise similarities. Assuming that we can represent this context dependency *directly* using collision probability $q(\cdot, \cdot)$, then the challenges discussed in importance sampling can be resolved. The coincidence of softmax probability $p(\cdot, \cdot)$ and LSH collision probability $q(\cdot, \cdot)$ makes $q(\cdot, \cdot)$ a sensible starting point for approximating self-attention. Specifically, to model dependency based on similarity, the collision probability aligns well with the exponential function in softmax in the domain of interest $[-1, 1]$ in Figure 1: both functions have positive zeroth, first and second order derivatives.

Note that (a) positive zeroth order derivative indicates that the dependency is positive, (b) positive first order derivative ensures that the dependency based on similarity is monotonic, and (c) positive second order derivative means that low similarity corresponds to almost no dependency. This leads us to hypothesize that a collision-based self-attention may be as effective as softmax-based self-attention. It can be formulated as,

$$\sum_{i=1}^{n} B_i(\boldsymbol{q}, \boldsymbol{k}_i)\boldsymbol{v}_i \qquad (3)$$

where $B_i(\boldsymbol{q}, \boldsymbol{k}_i)$ is a Bernoulli ran-

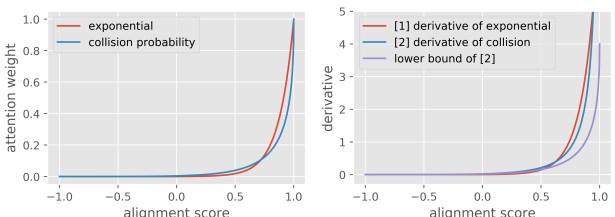

Figure 1: We compare attention weights using $\exp(\tau(x-1))$ with the collision probability of concatenating $\tau$ hyperplane hashes (Charikar, 2002) $(1 - \arccos(x)/\pi)^{\tau}$ for $\tau = 8$. We plot $\exp(\tau(x-1))$ so that the range is between 0 and 1 but without changing the actual attention weights in softmax. We also plot the derivative of exponential function and of collision probability, as well as a lower bound we will use later during backpropagation. Our method can be viewed as using LSH collision probability to estimate a biased approximation of exponential function.

dom variable where the success probability is given by the collision probability of $\boldsymbol{q}$ with the keys $\boldsymbol{k}_i$. Hence, it can be determined by the similarity between $\boldsymbol{q}, \boldsymbol{k}_i$. In a single hash, each $B_i(\boldsymbol{q}, \boldsymbol{k}_i)$ generates a realization to determine whether the corresponding token will be part of attention output or not. Conceptually, when sampling from softmax distribution, only one token is sampled as the attention output. In contrast, Bernoulli sampling determines whether each individual token is a part of the attention output. In principle, to determine the context dependency among tokens, you only need to sample once (YOSO) using a single hash to generate realizations of all Bernoulli random variables, $B_i(\boldsymbol{q}, \boldsymbol{k}_i), i = 1, \ldots, n$. Specifically, when keys are hashed to a hash table using a single hash, the realization of $B_i(\boldsymbol{q}, \boldsymbol{k}_i)$ for each query $\boldsymbol{q}$ will be 1 if $\boldsymbol{q}$ collides with $\boldsymbol{k}_i$, otherwise it will be 0. To our knowledge, using LSH collision probability to replace softmax dependencies for self-attention has not been studied before.

**YOSO-Attention**. By replacing softmax dependency with Bernoulli random variables and using LSH as an efficient sampler to estimate the success probability, we achieve an efficient self-attention (YOSO-Attention) to approximate softmax-based self-attention.

$$\text{YOSO}(\boldsymbol{Q}, \boldsymbol{K}, \boldsymbol{V}) = \mathcal{B}(\boldsymbol{Q}, \boldsymbol{K})\boldsymbol{V}; \quad \mathbb{E}[\text{YOSO}(\boldsymbol{Q}, \boldsymbol{K}, \boldsymbol{V})] = \left(1 - \frac{\arccos(\boldsymbol{Q}\boldsymbol{K}^T)}{\pi}\right)^{\tau}\boldsymbol{V} \qquad (4)$$

where $\mathcal{B}(\boldsymbol{Q}, \boldsymbol{K})$ is the Bernoulli sampling matrix using $m$ hashes.

$$\mathcal{B}(\boldsymbol{Q}, \boldsymbol{K})_{i,j} = \frac{1}{m}\sum_{k=1}^{m} \mathbb{1}_{f_k(\boldsymbol{Q}_{i,:})=f_k(\boldsymbol{K}_{j,:})} \quad \text{where } f_k, k = 1, \ldots, m \text{ are hash functions.} \qquad (5)$$

**Normalizing Attention**. In standard self-attention, each row of the softmax matrix is normalized so that the dependencies sum up to 1. In the above, we have discussed how the pairwise query-key dependencies can be estimated using Bernoulli sampling. We now present how to normalize the dependency in our method as standard self-attention. We can first estimate the dependencies and then normalize them using the sum of estimated dependencies estimated by $\mathcal{B}(\boldsymbol{Q}, \boldsymbol{K})\mathbf{1}$ where $\mathbf{1}$ is a vector of all entries being 1. $\mathcal{B}(\boldsymbol{Q}, \boldsymbol{K})\mathbf{1}$ can be computed by Eq. 4 by plugging $\mathbf{1}$ into $\boldsymbol{V}$. To make the estimation of self-attention more efficient, we turn to adopt a $\ell_2$ normalization to the attention output, similar as Levy et al. (2015) to use $\ell_2$ normalization for word embedding. Thus, attention outputs are invariant of the scaling, $\mathcal{B}(\boldsymbol{Q}, \boldsymbol{K})\mathbf{1}$, under $\ell_2$ normalization. Therefore, we have,

$$\text{N-YOSO}(\boldsymbol{Q}, \boldsymbol{K}, \boldsymbol{V}) = \ell_2(\mathcal{B}(\boldsymbol{Q}, \boldsymbol{K})\boldsymbol{V}) \qquad (6)$$

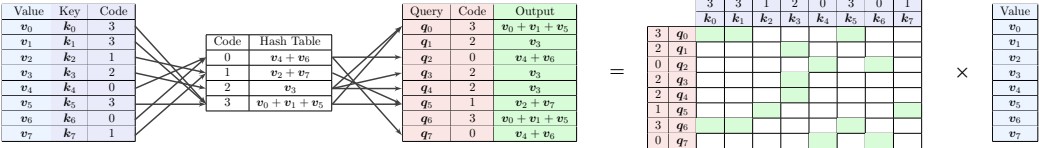

Figure 2: Overview of YOSO-Attention. The hash table stores the sum of values associated with hashed keys.

Empirically, we show the $\ell_2$ normalization does not affect the performance of our method as expected, which can be seen in Figure 3.

**LSH-based Bernoulli Sampling**. Now we discuss how to implement the procedure of using Bernoulli sampling to approximate self-attention. While a standard LSH procedure can be used, maintaining hash tables to store keys is inefficient on a GPU – the GPU memory size required for hash table cannot be predetermined and the workload might be skewed due to skewed bucket sizes. To tackle this issue, we propose LSH-based Bernoulli Sampling by only saving the summation of values corresponding to hashed keys instead of storing a collection of hashed keys.

The overview of our algorithm is shown in Figure 2. To compute $Y = \mathcal{B}(\boldsymbol{Q}, \boldsymbol{K})\boldsymbol{V}$, the procedure proceeds as follows. For each $k \in [1, \dots, m]$, we sample a hash function $f_k$ and create a hash table $\boldsymbol{H}^k \in \mathbb{R}^{2^\tau \times d}$ representing $2^\tau$ $d$-dimensional buckets. For each key $\boldsymbol{K}_{j,:}$, we add the value $\boldsymbol{V}_{j,:}$ to the bucket whose index is hash code $f_k(\boldsymbol{K}_{j,:})$, denoted as $\boldsymbol{H}^k_{f_k(\boldsymbol{K}_{j,:})}$,

$$\boldsymbol{H}^k_{f_k(\boldsymbol{K}_{j,:})} \leftarrow \boldsymbol{H}^k_{f_k(\boldsymbol{K}_{j,:})} + \boldsymbol{V}_{j,:} \tag{7}$$

Note that the size of $\boldsymbol{H}^k$ is $O(2^\tau d)$ and is independent of which bucket keys are hashed. With all keys processed for $k \in [1, \dots, m]$, for each query $\boldsymbol{Q}_{i,:}$, we maintain an output vector $\boldsymbol{Y}_{i,:}$ initialized to 0. Then, we allocate the bucket in $\boldsymbol{H}^k$ using $f_k(\boldsymbol{Q}_{i,:})$ for $k \in [1, \dots, m]$ and add all corresponding results in buckets to the output vector $\boldsymbol{Y}_{i,:}$ as

$$\boldsymbol{Y}_{i,:} \leftarrow \boldsymbol{Y}_{i,:} + \boldsymbol{H}^k_{f_k(\boldsymbol{Q}_{i,:}),:} \tag{8}$$

Therefore, each final output $\boldsymbol{Y}_{i,:}$ can be computed as,

$$\boldsymbol{Y}_{i,:} = \sum_{k=1}^{m} \sum_{j=1}^{n} \mathbb{1}_{f_k(\boldsymbol{Q}_{i,:})=f_k(\boldsymbol{K}_{j,:})} \boldsymbol{V}_{j,:} = \sum_{j=1}^{n} \mathcal{B}(\boldsymbol{Q}, \boldsymbol{K})_{i,j} \boldsymbol{V}_{j,:} \tag{9}$$

*Remark 3.* The memory and time complexity of this algorithm are $O(m2^\tau d)$ and $O(nmd)$ respectively, In addition, both time and memory are independent of the size of hash buckets. Further, We can improve the memory complexity to $O(m2^\tau)$ by reusing hash table and processing a few dimensions each time without increasing time complexity. The constant $\tau$ is small as it controls the decay rate of attention weight with respect to the angular distance between query and key, and it can be chosen to be a function of $\log_2(n)$. In our experiments, $\tau$ is set to $\log_2(n)$.

**Speed-up**. While not essential, we find that a fast random projection for computing the LSH hash code will be beneficial, since this step takes a large portion of the overall runtime. As suggested by Andoni et al. (2015), we use the approximated random projection to reduce time complexity to $O(nm\tau \log_2(d))$, allowing fast computation of hash codes.

**Backpropagation through YOSO-Attention**. For training, we also need to show backward propagation steps for YOSO-Attention. Here, we discuss this last component of YOSO-Attention which enables end-to-end and efficient training.

For backpropagation, the gradient of the loss $L$ w.r.t. $\boldsymbol{V}$ can be estimated similar to equation 4,

$$\nabla_{\boldsymbol{V}} L = ((1 - \frac{\arccos(\boldsymbol{Q}\boldsymbol{K}^T)}{\pi})^\tau)^T (\nabla_{\text{YOSO}} L) \approx \mathcal{B}(\boldsymbol{K}, \boldsymbol{Q})(\nabla_{\text{YOSO}} L) \tag{10}$$

The gradients of $L$ w.r.t. $\boldsymbol{Q}, \boldsymbol{K}$ are similar, so we only provide the expression for $\boldsymbol{Q}$,

$$\nabla_{\boldsymbol{Q}} L = \left( (\nabla_{\text{YOSO}} L) \boldsymbol{V}^T \right) \odot \left( \tau (1 - \frac{\arccos(\boldsymbol{Q}\boldsymbol{K}^T)}{\pi})^{\tau-1} \right) \oslash \left( \pi \sqrt{1 - (\boldsymbol{Q}\boldsymbol{K}^T)^2} \right) \boldsymbol{K} \tag{11}$$

where $\oslash, \odot$ are element-wise division and multiplication. The problem with the true gradient is that it goes to infinity as the alignment score between the query and the key approaches 1, which might lead to divergence. To avoid this numerical issue, we use a lower bound of the actual derivative of the collision probability, $[[(\nabla_{\text{YOSO}} L) \boldsymbol{V}^T] \odot \frac{\tau}{2}(1 - \frac{\arccos(\boldsymbol{Q}\boldsymbol{K}^T)}{\pi})^\tau] \boldsymbol{K}$, see Figure 1, which can be efficiently estimated via a variation of LSH-based Bernoulli Sampling. Specifically, note that the approximation can be decomposed into sum of $d$ LSH-based Bernoulli Sampling,

$$(\hat{\nabla}_{\boldsymbol{Q}} L)_{i,:} = \sum_{l=1}^{d} (\nabla_{\text{YOSO}} L)_{i,l} \sum_{j=1}^{n} \mathcal{B}(\boldsymbol{Q}, \boldsymbol{K})_{i,j} (\boldsymbol{V}_{j,l} \frac{\tau}{2} \boldsymbol{K}_{j,:}) \tag{12}$$

Therefore, following LSH-based Bernoulli Sampling, the memory complexity is $O(m2^\tau d^2)$, and time complexity is $O(nmd^2)$. The $d^2$ term can be eliminated by repeatedly using the same hash tables $d^2$ times without increasing runtime, which improves the memory complexity to $O(m2^\tau)$. The overall complexity of our method and comparison to standard self-attention is summarized in Table 1. Further, to address the quadratic dependence on $d$, in the Appendix, we will discuss a scheme to estimate the same quantity but is linear in $d$.

## 5 RELATED WORKS

There are a number of efforts describing ways to reduce the quadratic cost of self attention w.r.t. input sequence length. Among these works, Linformer (Wang et al., 2020) suggests that low rank attention might be sufficient and adds linear projections (on the sequence) to fixed size keys and values. There are also other low rank approximation ideas (Katharopoulos et al., 2020), (Choromanski et al., 2020) using separable functions on queries and keys to replace softmax self-attention. By assuming the self-attention rank to be independent of input sequence length, these methods can achieve $O(n)$ time and memory complexity. Another direction is to exploit the sparsity of softmax matrix and focus on certain sparsity patterns by only computing softmax dependencies within those patterns, including Sparse Transformer (Child et al., 2019), Longformer (Beltagy et al., 2020), and Big Bird (Zaheer et al., 2020) and Reformer (Kitaev et al., 2020). Note that, instead of using LSH as a tool to approximate nearest neighbor search to dynamically determine the sparsity pattern in Reformer, our YOSO-attention takes advantage of the connection of query-key similarity to the LSH collision probability to model the dependency among tokens.

## 6 EXPERIMENTS

In this section, we provide the empirical results for the proposed approach. To evaluate our proposed method, we follow the BERT language model pretraining procedure (Devlin et al., 2018) and evaluate the performance of our method in both intrinsic tasks and multiple downstream tasks in GLUE benchmark as well as runtime and memory relative to standard self attention. Previously, we assumed that queries and keys are unit length and described the construction to make it work. In the experiments, we found that simply applying a $\ell_2$ normalization on queries and keys and using a temperature $\tau$ as a hyperparameter does not degrade the performance of model and yet is more efficient to compute, so we use the simpler version in the experiments.

**BERT Pretraining**. Following Devlin et al. (2018), the model is pretrained on BookCorpus (Zhu et al., 2015) and English Wikipedia. To evaluate the capacity of model capturing the sentence level information, instead of using Next-Sentence-Prediction (NSP) as sentence level loss in the original BERT, we adapt the Sentence-Ordering-Prediction (SOP) proposed in ALBERT (Lan et al., 2019) as a more difficult task compared to NSP. All model are trained with Mask-Language-Modeling (MLM) and SOP objectives. We used the same hyperparameters for pretraining as Devlin et al. (2018). However, due to the computational resources limit, all models are trained for 500K steps. The batch size is set so that around $2^{17}$ tokens are processed per step. (batch size of 256 for sequence length 512, and batch size of 32 for sequence length 4096).

**Number of Hashes during Pretraining**. Since the estimation variance decreases as the number of hashes increases, to evaluate the trade-off between efficiency and performance in YOSO, we test on four hash settings: 16 hashes, 32 hashes, 64 hashes, and expectation of collision to simulate infinite hashes. We plot MLM validation perplexity and SOP validation loss curves of 512 length model pretrained with softmax self-attention and YOSO-Attention in the right plots of Figure 3. The curves of our method using expectation match and slightly exceed softmax self-attention, indicating our method is indeed as capable as self-attention. It is expected that as the number of hashes increase, the performance of our method will approach the curve using expectation as the approximation

| | Time | | Memory | |
|---|---|---|---|---|
| | Forward | Backward | Forward | Backward |
| Self-Attention | $O(n^2 d)$ | $O(n^2 d)$ | $O(n^2)$ | $O(n^2)$ |
| YOSO-Attention | $O(nm\tau \log_2(d) + nmd)$ | $O(nmd^2)$ | $O(nm\tau + m2^\tau)$ | $O(m2^\tau)$ |

Table 1: Time/memory complexity of self-attention and YOSO-attention in forward/backward computation

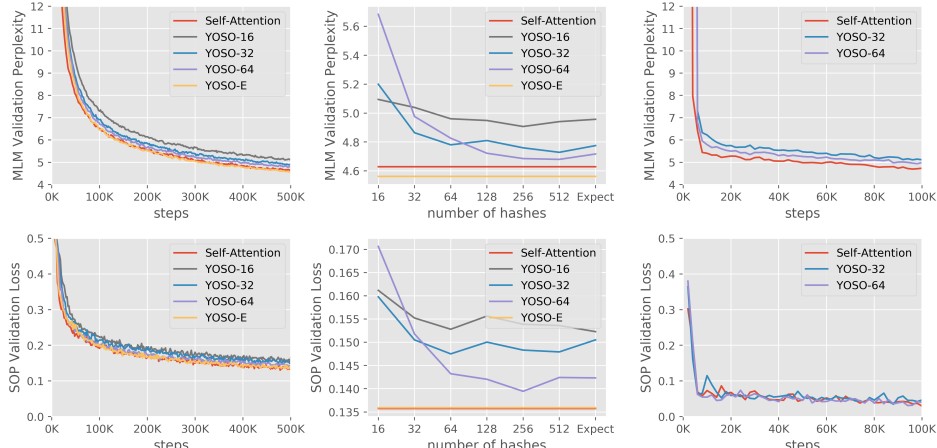

Figure 3: **(a)** The left two plots are results on MLM and SOP for 512 sequence length. We report MLM validation perplexity and SOP validation loss for each 2K training steps. **(b)** The middle two plots are results for MLM and SOP when using different number of hashes on validation. Since the runtime of YOSO-Attention is linear with respect to the number of hashes, these two plot directly reflect the equivalent relation between performance vs inference time. **(c)** The right two plots are results on on MLM and SOP for 4096 sequence length. YOSO-x means the model is pretrained with YOSO-Attention using x hashes with E being expectation.

become more accurate. For both MLM and SOP, we confirm that our method is as effective as softmax self-attention.

**Number of Hashes during Validation**. YOSO-Attention is a stochastic model. To make the inference deterministic, as in dropout (Srivastava et al., 2014), we take the expectation as our output. However, directly computing expectation involves a $O(n^2)$ cost, so we experiment with the effect of different hash settings in validation and simulate expectation as the number of hashes increases. We plot the MLM perplexity and SOP loss of the same pretrained models using different number of hashes on validation in the center plots of Figure 3. We observe that as the number of hash increases, the MLM perplexity and SOP loss generally decreases for all pretraining hash settings.

**Pretraining on Longer Sequence**. To examine whether our method can scale linearly with sequence length, we continue to pretrain BERT-base models using the corresponding 500K step checkpoints for 512 length model, and add additional positional embedding as suggested in Beltagy et al. (2020). We observe that compared to 512 sequence length, the small performance gap between YOSO-Attention and softmax self-attention does not increase as suggested in the left plots of Figure 3, providing evidence that the number of hashes can be chosen independent of sequence length.

**GLUE Tasks**. In addition to intrinsic tasks, we examined the effectiveness of our method on diverse downstream tasks and ask how our method compares with standard attention even after finetuning. We finetuned all pretrained BERT-base model on MRPC (Dolan & Brockett, 2005), RTE (Giampiccolo et al., 2007), SST-2 (Socher et al., 2013), QNLI (Rajpurkar et al., 2016), QQP (Chen et al., 2018), and MNLI

| Method | MRPC | RTE | SST-2 | QNLI | QQP | MNLI-m/mm |
|---|---|---|---|---|---|---|
| Self-Attention | 88.3 | 70.8 | 91.1 | 90.3 | 87.3 | 82.4/82.4 |
| YOSO-16 | 87.1 | 68.6 | 90.7 | 88.3 | 85.3 | 79.6/79.5 |
| YOSO-32 | 87.3 | 71.8 | 90.9 | 89.0 | 86.3 | 80.5/80.7 |
| YOSO-64 | 88.1 | 69.7 | 91.5 | 89.5 | 87.0 | 81.6/81.6 |
| YOSO-E | 88.1 | 74.4 | 92.3 | 90.1 | 87.3 | 82.2/82.9 |
| YOSO-16-E | 87.8 | 69.3 | 91.5 | 89.3 | 86.8 | 81.0/81.4 |
| YOSO-32-E | 87.8 | 70.4 | 91.1 | 90.1 | 86.8 | 80.8/81.4 |
| YOSO-64-E | 88.3 | 72.2 | 91.7 | 90.0 | 87.2 | 81.9/82.8 |

Table 2: Dev set results on GLUE tasks. We report F1 score for MRPC and QQP and accuracy for others. YOSO-x means the same as in Figure 3 and YOSO-x-E means that YOSO-x is finetuned on downstream tasks using expectation.

(Williams et al., 2018) tasks in the GLUE benchmarks and report their corresponding dev metrics. For large datasets including QNLI, QQP, and MMNL, due to extensive resource needs, we cannot do hyperparameter search, so we used a batch size of 32 and learning rate 3e-5 to update our model and finetune our models for 4 epochs. For MRPC, RTE, and SST-2, we follow BERT finetuning to do a hyperparameter search with candidate batch size {8, 16, 32} and learning rate {2e-5, 3e-5, 4e-5, 5e-5} and select the best dev set result. Results are listed in Table 2. We observed that YOSO's performance on downstream tasks is comparable with standard attention, and even has slightly better

results in some hash settings. Further, the downstream performance of YOSO generally increases as more hashes are used, providing an adjustable trade-off between efficiency and accuracy.

**Longer Sequence Task**. To further evaluate YOSO on long sequence tasks, we extended the positional embeddings of a trained YOSO-64 model and used it as an initialization to train a 4096 length YOSO-128 model using a batch size of 64 and learning rate 5e-5 on BookCorpus (Zhu et al., 2015), English Wikipedia, one third of the Stories (Trinh & Le, 2018), and one third of Realnews (Zellers et al., 2019) for 100K steps, similar to Longformer pretraining (Beltagy et al., 2020). Then, we finetuned our model on WikiHop (Welbl et al., 2018). Due to the computational resource limits, we only tested a small set of hyperparameters (batch size = 32, learning rate $\in$ {1e-5, 2e-5, 4e-5}, number of epochs = 10). The dev accuracy is 73.7 for YOSO-128-E, which is comparable to 73.8 in Longformer-512 (see caption in Table 3) without hyperparameter search but slightly worse than 75.0 that Longformer-512 achieves with hyperparameter search.

**Comparisons to Baselines**. Apart from comparing YOSO to standard self-attention, we also evaluated its competitiveness with other efficient attention methods. To keep the financial costs of these experiments reasonable, instead of training all methods from scratch, we used RoBERTa-base's pretrained weights as the starting point and trained each model using batch size 512 and learning rate 5e-5 on BookCorpus (Zhu et al., 2015) and English Wikipedia for 95K steps. Then, we finetuned the models on SST-2, QQP, and MNLI. These results are shown in Table 3. We observed that our performance is competitive with other baselines while the memory consumption of YOSO is much less ($2.6\times$, $1.9\times$, $2.1\times$ memory savings compared to Reformer, Longformer, and Linformer respectively, see Backward-Cache in Table 4). This has potential ramifications for training

| Method | SST-2 | QQP | MNLI-m/mm |
|---|---|---|---|
| YOSO-32 | 93.5 | 87.3/90.5 | 84.4/84.1 |
| YOSO-64 | 94.2 | 87.9/90.9 | 85.1/85.2 |
| Reformer-2 | 92.9 | 87.8/91.0 | 85.6/85.3 |
| Longformer-64 | 94.9 | 88.4/91.4 | 85.3/85.2 |

Table 3: Dev set results on SST-2, QQP, and MNLI. We report both F1 score and accuracy QQP and accuracy for others. Reformer-x: Reformer using HuggingFace implementation (Wolf et al., 2019) using x hashes. Longformer-x: Longformer using HuggingFace implementation (Wolf et al., 2019) using sliding window of size x. YOSO-x is similar to description in Figure 3. We did not re-train Linformer to keep compute costs reasonable; when trained from scratch Linformer (Wang et al., 2020) reports accuracy 93.4 for SST-2 and 90.8 for QQP.

such models with more moderate hardware resources which are much less expensive. Further, notice that YOSO is potentially applicable to a wider range of applications, especially where the input sequence represents an unordered set of high dimensional points (where spatial locality of the input sequence may not hold).



Figure 4: Attention matrices generated by self-attention and YOSO-Attention with different hash settings using the same input. Notice that the patterns are preserved well.

**Estimation Error**. To assess the effectiveness of our algorithm, using $\boldsymbol{Q}, \boldsymbol{K}$ from the trained model, we generated attention matrices using our algorithm with different number of hashes and compare it against standard self-attention. In Figure 4, visually, we see that our method produces similar attention patterns as standard self-attention. The estimation of attention matrix becomes more accurate as the number of hashes increases. Further, each output of YOSO-Attention is a weighted sum of random variables as shown in equation 3; so one may suspect that as the sequence length increases, the variance of YOSO-Attention output might potentially increase. We did not observe this behavior which may be partly due to the hyperparameter $\tau = O(\log(n))$ that controls the decay rate of LSH collision probability as the similarity changes. We can also ask whether or not the estimation error of YOSO-Attention for a fixed number of hashes increases as the sequence length increases. We use $\boldsymbol{Q}, \boldsymbol{K}, \boldsymbol{V}$ generated by the pretrained model and estimate the error between N-YOSO($\boldsymbol{Q}, \boldsymbol{K}, \boldsymbol{V}$) and $\mathbb{E}[\text{N-YOSO}(\boldsymbol{Q}, \boldsymbol{K}, \boldsymbol{V})]$. As the left plot of Figure 5 suggests, the relative error of our method stays almost constant as the sequence length increases from 128 to 4096. This indicates that using sampling to estimate attention weight based on YOSO-Attention can scale up with sequence length and preserve the same estimation quality without increasing the number of hashes.

**Runtime and Memory**. We measure the runtime of our method as sequence length increases. To show the trend more precisely, we measured the runtime per token as shown in Figure 5 (right). There is a slight increase in runtime per token as the sequence length increases, but note that the $x$-axis of the plot is log scale, so the increment is small. When the sequence length increases by $32\times$, the runtime per token only increases by $30\%$, which is explained by our choice of hyperparameter $\tau = O(\log(n))$. Aside from the plot, we report the training and testing efficiency of our method as well as three other efficient attention meth-

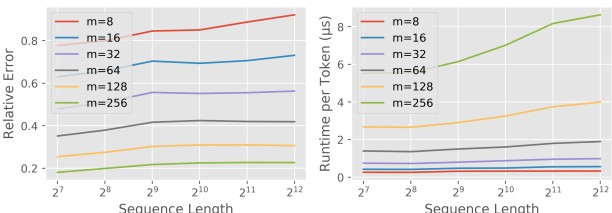

Figure 5: **(a)** The relative error in the left plot is defined as $\mathbb{E}\big[\frac{\|\mathbb{E}[\text{N-YOSO}(\boldsymbol{Q},\boldsymbol{K},\boldsymbol{V})]-\text{N-YOSO}(\boldsymbol{Q},\boldsymbol{K},\boldsymbol{V})\|_\infty}{\|\mathbb{E}[\text{N-YOSO}(\boldsymbol{Q},\boldsymbol{K},\boldsymbol{V})]\|_\infty}\big]$. The relative error is estimated by computing $\mathbb{E}[\text{N-YOSO}(\boldsymbol{Q},\boldsymbol{K},\boldsymbol{V})]$ based on collision probability, then estimating N-YOSO$(\boldsymbol{Q},\boldsymbol{K},\boldsymbol{V})$ multiple times, finally computing the mean of relative error of each estimate as an estimation of the outer expectation. **(b)** The runtime per token is estimated by estimating N-YOSO$(\boldsymbol{Q},\boldsymbol{K},\boldsymbol{V})$ multiple times and measuring the total time elapsed and then dividing the total time by number of iterations and sequence length to get runtime per token.

ods against standard self-attention. The results were measured using $\boldsymbol{Q},\boldsymbol{K},\boldsymbol{V}$ of a specified sequence length generated by a trained model and fed into a BERT-base Multi-Head Attention module multiple times. The experiments were performed on a single NVIDIA 2080TI. From Table 4, we can see that while for a standard $512$ length sequence, our method has a similar runtime as self-attention, as the sequence length increases, the speed-up and memory savings become significant. While our method offers similar runtime savings as other efficient attention methods, the memory consumption for training (i.e., Backward-Cache) of our method is much lower than all other methods in almost all settings.

| | Sequence Length | | | | | | | |
|---|---|---|---|---|---|---|---|---|
| Model | 512 | 1024 | 2048 | 4096 | 512 | 1024 | 2048 | 4096 |
| | Forward / Backward Time (ms) | | | | Backward-Cache / Training-Peak / Testing-Peak Memory (MB) | | | |
| Self-Attention | 0.6 / 0.88 | 1.81 / 2.29 | 5.92 / 8.04 | 21.2 / 29.73 | 51 / 70 / 48 | 179 / 260 / 173 | 671 / 999 / 659 | 2589 / 3918 / 2565 |
| YOSO-16 | 0.54 / 0.93 | 1.09 / 1.87 | 2.29 / 3.73 | 4.6 / 7.38 | 17 / 32 / 27 | 34 / 63 / 55 | 68 / 127 / 110 | 136 / 253 / 243 |
| YOSO-32 | 0.71 / 1.21 | 1.49 / 2.43 | 3.11 / 4.88 | 6.31 / 9.89 | 18 / 44 / 40 | 36 / 89 / 80 | 71 / 178 / 161 | 142 / 355 / 345 |
| YOSO-64 | 1.09 / 1.91 | 2.26 / 3.9 | 4.87 / 7.84 | 10.21 / 16.02 | 19 / 70 / 66 | 39 / 140 / 131 | 77 / 279 / 263 | 154 / 559 / 549 |
| YOSO-128 | 1.91 / 3.17 | 4.09 / 6.45 | 9.16 / 13.37 | 19.41 / 28.85 | 22 / 121 / 117 | 45 / 242 / 233 | 89 / 483 / 467 | 178 / 967 / 957 |
| Linformer-128 | 0.42 / 0.51 | 0.86 / 1.03 | 1.72 / 2.16 | 3.53 / 4.76 | 25 / 33 / 17 | 50 / 66 / 35 | 101 / 136 / 71 | 166 / 247 / 154 |
| Linformer-256 | 0.53 / 0.67 | 1.05 / 1.36 | 2.16 / 2.85 | 4.02 / 6.16 | 38 / 55 / 24 | 76 / 111 / 49 | 156 / 233 / 102 | 239 / 416 / 227 |
| Reformer-2 | 0.73 / 1.14 | 1.52 / 2.29 | 3.09 / 4.59 | 6.35 / 9.19 | 48 / 61 / 38 | 96 / 122 / 76 | 192 / 243 / 151 | 384 / 487 / 302 |
| Reformer-4 | 1.22 / 2.05 | 2.47 / 4.1 | 5.03 / 8.19 | 10.49 / 16.36 | 86 / 114 / 71 | 172 / 228 / 143 | 344 / 455 / 286 | 688 / 911 / 572 |
| Reformer-8 | 2.17 / 3.85 | 4.44 / 7.69 | 9.04 / 15.4 | 18.84 / 30.83 | 162 / 220 / 139 | 324 / 439 / 278 | 647 / 878 / 555 | 1294 / 1756 / 1111 |
| Longformer-64 | 0.81 / 1.71 | 1.63 / 3.37 | 3.28 / 6.87 | 6.19 / 13.67 | 35 / 43 / 25 | 71 / 86 / 50 | 142 / 172 / 99 | 224 / 307 / 198 |
| Longformer-128 | 0.9 / 2.07 | 1.81 / 4.12 | 3.62 / 8.41 | 6.89 / 16.51 | 41 / 48 / 32 | 82 / 96 / 64 | 165 / 193 / 127 | 269 / 345 / 253 |
| Longformer-256 | 1.09 / 2.74 | 2.19 / 5.56 | 4.45 / 10.89 | 8.47 / 22.91 | 51 / 64 / 46 | 104 / 129 / 91 | 208 / 258 / 182 | 360 / 481 / 363 |
| Longformer-512 | 1.42 / 3.95 | 2.96 / 8.65 | 6.05 / 17.86 | 11.54 / 36.14 | 72 / 103 / 80 | 148 / 211 / 159 | 299 / 423 / 317 | 538 / 767 / 585 |

Table 4: The reported values pertain to a single instance. Time is estimated by averaging total runtime and then dividing it by batch size, while memory is measured by dividing total memory consumption by batch size. Linformer-x: Linformer model where the sequence is projected to a fixed $x$ length. The rest of method names are consistent with Table 3. Backward-Cache: amount of memory required to cache certain tensors for backward pass (e.g., activation and inputs which is required for gradient calculation or is cached to avoid redundant calculation). Since the Backward-Cache will accumulate as model grows deeper, this part is usually the most significant portion of memory consumption. Training-Peak: maximum memory needed during training including temporary tensors and cached tensors for backward pass. Some tensors are not required for backward pass, and they are only held in memory temporarily. As model grows deeper, the amount of memory for temporary tensors does not grow. Testing-Peak: Similar to Training-Peak, but nothing is cached for backward pass.

## 7  CONCLUSION

We presented a transformer-based model, YOSO-Attention, that scales linearly in the number of input tokens. This allows the model to be applicable to a wide range of long document NLP tasks. Via a randomized sampling based scheme, our model approximates self-attention as a sum of individual tokens associated with Bernoulli random variables that can be sampled at once by a single hash, in principle. With specific modifications of LSH, YOSO-Attention can be efficiently deployed within a deep learning framework and various aspects of this idea and our implementation, we expect, will find use in other novel settings and applications (e.g., in vision).

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

# A APPENDIX

In Appendix, we provide some details of our method that are left out in the main text.

**Backpropogation Derivation** When using expectation of LSH collision as attention weights, the attention of one query $\boldsymbol{q}$ to keys $\boldsymbol{k}^i$ and associated values $\boldsymbol{v}^i$ for all $i \in \{1, ..., n\}$ is defined as

$$\boldsymbol{y} = \sum_{i=1}^{n} \left(1 - \frac{\arccos(\boldsymbol{q}^T \boldsymbol{k}^i)}{\pi}\right)^\tau \boldsymbol{v}^i \tag{13}$$

then we want to compute the gradient of loss w.r.t. $\boldsymbol{q}$, which we denoted as $\nabla_{\boldsymbol{q}} L$, with the gradient of loss w.r.t. $\boldsymbol{y}$ denoted, $\nabla_{\boldsymbol{y}} L$, given. We start by computing the $p$-th entry of $\nabla_{\boldsymbol{q}} L$:

$$\frac{\partial L}{\partial \boldsymbol{q}_p} = \sum_{j=1}^{d} \frac{\partial L}{\partial \boldsymbol{y}_j} \frac{\partial \boldsymbol{y}_j}{\partial \boldsymbol{q}_p} = \sum_{j=1}^{d} \frac{\partial L}{\partial \boldsymbol{y}_j} \frac{\partial}{\partial \boldsymbol{q}_p} \left[ \sum_{i=1}^{n} \left(1 - \frac{\arccos(\boldsymbol{q}^T \boldsymbol{k}^i)}{\pi}\right)^\tau \boldsymbol{v}_j^i \right] \tag{14}$$

Then use $\frac{d}{dx}(1 - \frac{\arccos(x)}{\pi})^\tau = \frac{\tau(1 - \frac{\arccos(x)}{\pi})^{\tau-1}}{\pi\sqrt{1-x^2}}$ and plug it into the Eq. 13

$$\frac{\partial L}{\partial \boldsymbol{q}_p} = \sum_{j=1}^{d} \frac{\partial L}{\partial \boldsymbol{y}_j} \sum_{i=1}^{n} \left( \frac{\tau \left(1 - \frac{\arccos(\boldsymbol{q}^T \boldsymbol{k}^i)}{\pi}\right)^{\tau-1}}{\pi\sqrt{1 - (\boldsymbol{q}^T \boldsymbol{k}^i)^2}} \boldsymbol{k}_p^i \right) \boldsymbol{v}_j^i \tag{15}$$

After swapping the order of two summations, Eq. 13 becomes

$$\frac{\partial L}{\partial \boldsymbol{q}_p} = \sum_{i=1}^{n} (\nabla_{\boldsymbol{y}} L)^T \boldsymbol{v}^i \frac{\tau \left(1 - \frac{\arccos(\boldsymbol{q}^T \boldsymbol{k}^i)}{\pi}\right)^{\tau-1}}{\pi\sqrt{1 - (\boldsymbol{q}^T \boldsymbol{k}^i)^2}} \boldsymbol{k}_p^i \tag{16}$$

Note that only $\boldsymbol{k}_p^i$ is different for different entries of $\nabla_{\boldsymbol{q}} L$, so we can write it as

$$\nabla_{\boldsymbol{q}} L = \sum_{i=1}^{n} (\nabla_{\boldsymbol{y}} L)^T \boldsymbol{v}^i \frac{\tau \left(1 - \frac{\arccos(\boldsymbol{q}^T \boldsymbol{k}^i)}{\pi}\right)^{\tau-1}}{\pi\sqrt{1 - (\boldsymbol{q}^T \boldsymbol{k}^i)^2}} \boldsymbol{k}^i \tag{17}$$

Equation 11 is the matrix form of above

$$\nabla_{\boldsymbol{Q}} L = \left( \left((\nabla_{\text{YOSO}} L) \boldsymbol{V}^T\right) \odot \left(\tau(1 - \frac{\arccos(\boldsymbol{Q}\boldsymbol{K}^T)}{\pi})^{\tau-1}\right) \oslash \left(\pi\sqrt{1 - (\boldsymbol{Q}\boldsymbol{K}^T)^2}\right) \right) \boldsymbol{K} \tag{18}$$

Note that $\pi\sqrt{1 - (\boldsymbol{Q}\boldsymbol{K}^T)^2}$ approaches to 0 as alignment score between the query and the key approaches 1, so we use the fact that $\frac{1}{2}(1 - \frac{\arccos(x)}{\pi}) \leq \frac{1}{\pi\sqrt{1-x^2}}$ for $x \in [-1, 1]$ and define a lower bound to replace the actual gradient

$$\nabla_{\boldsymbol{Q}} L = \left( \left((\nabla_{\text{YOSO}} L) \boldsymbol{V}^T\right) \odot \left(\frac{\tau}{2}(1 - \frac{\arccos(\boldsymbol{Q}\boldsymbol{K}^T)}{\pi})^\tau\right) \right) \boldsymbol{K} \tag{19}$$

**Approximating Random Projection in LSH**. In the main text, we discussed how to estimate self-attention using Bernoulli sampling via LSH. The first step of using LSH is computing hash code using random projection. To compute hash codes for a vector $\boldsymbol{x}$, we proceed as follows.

$$F : \mathbb{R}^d \to \{0, 1\}^{m\tau} \quad F(\boldsymbol{x}) = \text{sign}(\mathbf{R}\boldsymbol{x}) \tag{20}$$

where $\mathbf{R} \in \mathbb{R}^{(m\tau) \times d}$, $\mathbf{R}_{ij} \sim \mathcal{N}(0, 1)$, then the output vector are partition to $m$ $\tau$-dimensional binary hash code. The time complexity for random project is $O(nm\tau d)$. To efficiently approximate random projection, we follow the construction used in Andoni et al. (2015). The output of $m\tau$-dimensional vector is divided to $\frac{m\tau}{d}$ $d$-dimensional vectors, then hash codes are estimated by

$$F(\boldsymbol{x}) = \text{concat}(\text{sign}(\boldsymbol{H}\boldsymbol{D}_3^1\boldsymbol{H}\boldsymbol{D}_2^1\boldsymbol{H}\boldsymbol{D}_1^1\boldsymbol{x}), ..., \text{sign}(\boldsymbol{H}\boldsymbol{D}_3^{\frac{m\tau}{d}}\boldsymbol{H}\boldsymbol{D}_2^{\frac{m\tau}{d}}\boldsymbol{H}\boldsymbol{D}_1^{\frac{m\tau}{d}}\boldsymbol{x})) \tag{21}$$

where $\boldsymbol{D}_i^j$ are diagonal matrices with entries uniformly sampled from $\{-1, +1\}$, and $\boldsymbol{H}$ is Hadamard matrix. This approximation reduce time complexity to $O(nm\tau \log_2(d))$.

**Alternative Procedure for Approximating Backpropagation**. In the main text, we provided a procedure as shown in Eq. 12, which use LSH-based Bernoulli Sampling $d$ times as subroutine. The complexity of this procedure is linear w.r.t. sequence length $n$, which is desirable but the runtime can be large if $d$ is relatively large. Therefore, we provide second procedure, which is linear with respect to $d$. The gradient of $L$ w.r.t. the $i$-th row of $\boldsymbol{Q}$ is written as

$$(\hat{\nabla}_{\boldsymbol{Q}}L)_{i,:} = \sum_{j=1}^{n}(\nabla_{\text{YOSO}}L)_{i,:}^{T}\boldsymbol{V}_{j,:}\mathcal{B}(\boldsymbol{Q},\boldsymbol{K})_{i,j}\frac{\tau}{2}\boldsymbol{K}_{j,:} \tag{22}$$

Note that if $\mathcal{B}(\boldsymbol{Q},\boldsymbol{K})_{i,j}$ is zero then the corresponding summation term does not need to be computed. The alternative procedure counts the number of success in $m$ samples at each entry $\mathcal{B}(\boldsymbol{Q},\boldsymbol{K})_{i,j}$ and only computes the summation term when $\mathcal{B}(\boldsymbol{Q},\boldsymbol{K})_{i,j}$ is non-zero, and thus the runtime is $O(\texttt{nnz}(S(A,B))(m+d))$ (counting number of success + computing nonzero terms). In the worst case, $\texttt{nnz}(\mathcal{B}(\boldsymbol{Q},\boldsymbol{K})) = n^2$, it would be as expensive as dense matrix multiplications in complexity and even worst in practice due to large memory latency resulted from indirect memory access. However, in practice, $\mathcal{B}(\boldsymbol{Q},\boldsymbol{K})$ is generally sparse if $\tau$ is set properly. Further, the first procedure guarantees a linear complexity scaling of our method for extremely long sequences. As an improvement, we can dynamically select one from these two method based on runtime, than the time complexity is $O(\min(nmd^2, \texttt{nnz}(\mathcal{B}(\boldsymbol{Q},\boldsymbol{K}))(m+d)))$.

