# OpenReview forum: "You Only Sample (Almost) Once: Linear Cost Self-Attention Via Bernoulli Sampling"
_ICLR.cc/2021/Conference — Reject_

### Official Review · AnonReviewer3 · 2020-10-27
**Lacking impact and convincing experiments, unnecessary model**

**Rating:** 2
**Confidence:** 4

**Review:**

This paper presents a linear-time attention model based on importance sampling and locality sensitive hashing. The idea is to use Bernoulli sampling to approximate the self-attention - which is quadratic in complexity.

Honestly, this is a very crowded space and already many models exist (https://arxiv.org/abs/2009.06732). The authors are aware of these works, cite them and yet there is no comparison.

This method seems to be rooted in LSH and a very natural question is how does compare to Reformers. There is not even a sparse transformer or local attention baseline in the experiments. This raises questions about whether this paper will even make any impact at all. (comparisons with longformer is done only on speed/memory but not qualitatively). why?

I also find other flaws with the model. If sampling is used, this essentially makes the model stochastic (correct me if Im wrong here). but there are undesirable properties of this such as having non-deterministic inference.

Another flaw is that method potentially introduces a lot of instability in training. I think the authors could comment a little on this. Transformers are already notoriously difficult to train and I figure that this method would probably make it way harder for practitioners to get the hyperparameters correct. I think playing with the scaling and normalization of the self-attention weights is something non-ideal, and unless the authors can show this is reasonable stable i am not convinced.

I also find it difficult to understand the choices of tasks. It seems like GLUE benchmark is used, yet most of the tasks (like SST) are relatively shorter sequences. I think the authors need to explore datasets that showcase the model's ability on longer sequences. Artificially raising sequence len during pretraining is not really sufficient to be convincing that the model is doing something useful for longer sequences (since the masked out tokens really depend on local context).

My constructive feedback to the authors to improve the paper is to have reasonable baselines for comparison. The datasets are also not appropriate. I would suggest some actually long-range tasks in order to showcase the model's capabilities.

At the rate of the number of new models that tackle this problem, I suspect it would be wise to wrap up your sleeves and add actual efficient transformer baselines.

---

> ### Author Response · Authors · 2020-11-18
> **Response to Reviewer 3**
>
> We thank the reviewer and answer the main questions below.
>
> -
>
> Q) Rooted in LSH, so how does it compare to Reformers. There is not even a sparse transformer or local attention baseline in the experiments.
>
> We hope that the reviewer agrees that Reformer, sparse attention or local attention all approximate softmax self-attention and few, if any, efficient self-attention methods outperform softmax self-attention by a non-trivial margin, so if we can show that our method is as good as softmax self-attention (see MLM and SOP in Figure 3 and GLUE in Table 2), the comparison should be applicable to other efficient self-attention methods as well. But we appreciate the comments. We have run experiments to compare to some other efficient attention methods and updated our paper with the experiment results (see Table 3).
>
> -
>
> Q) Comparisons with longformer only on speed/memory but not qualitatively. Why?
>
> The reason is related to the above answer. Longformer usually performs slightly worse/comparable to self-attention on many tasks (see Table 11 in Longformer where RoBERTa and Longformer (seqlen:512), see Figure 4 in Reformer, and Figure 3 and Table 2 in Linformer). A new algorithm performing as well as softmax self-attention is promising qualitative evidence. That leaves two unresolved issues regarding the effectiveness of the method: runtime and memory. To this end, we showed in Table 4 where the method shows promising benefits relative to both Linformer and Longformer (already in the paper) and Reformer (added in the revised version).  But we appreciate the suggestion: we have run experiments for qualitative comparisons (see Table 3).
>
> -
>
> Q) Flaws. If sampling is used, this essentially makes the model stochastic/non-deterministic inference.
>
> In general, the main issue with a generic (without any guarantees) stochastic procedure during testing time is that it is often not possible to know the impact of randomness on downstream task performance. Indeed, the popular regularization choice of Dropout had similar issues that was addressed by simply scaling outputs of layers appropriately. Intuitively, our l2 normalization in equation (6) performs this role for our YOSO module.
> We did not find any instability/inconsistency during testing time in our experiments at all. Even ignoring experimental evidence, our result in equation (4) indicates that the YOSO module can be seen as an Elementary Block as in Equation (1) (and Remark 3) in https://arxiv.org/pdf/1811.07429.pdf. Since LSH functions preserve distances approximately (by definition), they are (approximately) Lipschitz, which (in particular) implies that the YOSO module may be robust to input perturbations (Proposition 1 in https://arxiv.org/pdf/1811.07429.pdf).
> For these reasons and given the broad use of randomization in all phases of algorithm design in machine learning, we do not see a conceptual flaw, especially for natural language tasks considered here.
>
> -
>
> Q) Another flaw: potentially introduces a lot of instability in training. Comment a little on this.
>
> We assure the reviewer that this is not true. In all our experiments, training was extremely stable and never a concern at all. As seen in Figure 3, the model converges, even with 16 hashes. Further, the plot suggests that the convergence rate is similar to standard self-attention.
> If the reviewer suspects that the instability arises from hyperparameters, we note that during training,  all parameters (other than just *two* LSH parameters) are exactly like the original BERT paper.
>
> -
>
> Q) Difficult to understand the choices of tasks. It seems like GLUE benchmark is used, yet most of the tasks (like SST) are relatively shorter sequences.
>
> Our experimental focus was on language modeling, and it is not uncommon to test efficient attention with GLUE tasks. For example, Linformer solely relies on showing performance metrics on MLM pretraining and GLUE downstream tasks. Nevertheless, we appreciate the comment; we are now running experiments targeted specifically to measure long sequence modeling and will report results shortly.
>
> -
>
> Q) Artificially raising sequence len during pretraining is not really sufficient to be convincing that the model is doing something useful for longer sequences. Suggest some actually long-range tasks in order to showcase the model's capabilities.
>
> The reviewer can verify that this scheme is commonly used in other approaches also. In fact, Linformer (Wang et al., 2020) utilizes this approach (page 7, Figure 3 and Table 2) and shows MLM perplexity on different sequence lengths. Longfomer also reports MLM performance of their model using BPC on longer sequences (see Table 6 in page 7), and shows that as the sequence length increases, the MLM performance can actually increase slightly.
> Nonetheless, we understand the reviewer’s concern and results on longer sequences are forthcoming.

---

> ### Author Response · Authors · 2020-11-24
> **Experiments on Long Sequence Tasks**
>
> We thank the reviewers for their comments. Based on the suggestion of Reviewer 3, we finished experiments on WikiHop task that is explicitly designed to evaluate YOSO on longer sequence tasks. Due to limited time/resources, we could try a couple of hyperparameter settings. Nonetheless, the results are promising and consistent with the rest of the paper. This sub-section is now included in blue on page 8 (top). The dev accuracy of WikiHop task is 73.7 for YOSO-128-E (see caption about naming convention in Table 2), which is comparable to 73.8 in Longformer without hyperparameter search but slightly worse than 75.0 that Longformer achieves with hyperparameter search.

---

### Official Review · AnonReviewer1 · 2020-10-28
**Elegant idea and formulation but somewhat lacking evaluation**

**Rating:** 6
**Confidence:** 5

**Review:**

### Summary

The paper proposes to replace the weighted average of the values in standard self-attention with the average of values sampled in a way that the expectation is close to the result of self-attention. In particular, the authors associate with each query-key pair, a Bernoulli random variable with expected value close to the exponential of the dot-product. Sampling these variables and averaging the values per query is formulated in an efficient way using locality-sensitive hashing.

### Strengths

- Using sampling to approximate self-attention is novel and promising.
- The LSH formulation where the values are averaged in the bucket is a clever way to avoid the pairwise interactions between queries and keys.
- Training with hashing and evaluating using the expectation is interesting and provides evidence for the approximation quality of YOSO.

### Weaknesses

1. In the evaluation there is never an explicit comparison with respect to both time and performance. I appreciate that given a large enough sequence length YOSO will always be faster but will it be good enough?

2. One of the most important parts of the methodology, the gradient computation, is the least clearly written. For instance, equation 11 contains $\nabla_{Attn}$ which is never defined and subsequent equations contain $\nabla_{YOSO}$ which seems to contradict equation 11. Moreover, how is equation 11 derived? Is it the gradient of the expectation?

3. In table 3, the performance is only measured with respect to inference. Given that the most computationally intensive part of the method is the backward pass, a comparison with respect to wall-clock time per epoch, as well as total training time would be very informative.

### Reasons for recommendation

I find the idea very elegant and interesting however, the experimental section is somewhat lacking. There is no clear evaluation of the trade-off between speed and performance. The MLM task, although significant and demanding, contains sequences of small length, otherwise why not show a graph of performance vs inference-time.

---

> ### Author Response · Authors · 2020-11-18
> **Response to Reviewer 1**
>
> We thank the reviewer and answer the main questions below.
>
> -
>
> Q) In evaluation there is never an explicit comparison with respect to both time and performance. I appreciate that given a large enough sequence length YOSO will always be faster but will it be good enough?
>
> Ideally, we would want to train a range of models with different sequence lengths and different LSH settings (number of hashes), and evaluate them on the same tasks to compare time and performance of our method as a function of different sequence lengths and LSH settings. But the reviewer will see that the scale of experiments needed to generate this specific comparison plot will be infeasible unless the compute cycles available to us are practically free. Nonetheless, we did want to present evidence to showcase such type of behavior in a budget friendly form that a reader will still find valuable: we included a 4096 sequence length MLM and SOP task (the right two plots of Figure 3) to evaluate our algorithm on a long sequence task, which indeed shows the viability and benefits of our method.
> Currently, we are also running experiments on other long sequence tasks, and will report these results shortly. In addition, we also included in this revised version of the paper, a plot of relative estimation error of our method (Figure 5 in the Experiments). It shows that for a fixed number of hashes, the relative estimation error stays almost the same when the sequence length increases from 128 to 4096. This is promising and indicates that the strategy of using sampling to estimate attention weights can scale up with sequence length without increasing estimation error. This lends support to our key idea and the behavior is consistent with our findings reported throughout the paper.
>
> -
>
> Q) One of the most important parts of the methodology, the gradient computation, is the least clearly written. For instance, equation 11 contains $\nabla$Attn  which is never defined and subsequent equations contain $\nabla$YOSO which seems to contradict equation 11. Moreover, how is equation 11 derived? Is it the gradient of the expectation?
>
> Thanks for the suggestion. $\nabla$Attn is a typo, it should be $\nabla$YOSO. Regarding gradient calculation, the gradient w.r.t. Q and K is calculated based on expectation. We have updated the text, and included a detailed section of how the gradient is calculated in the Appendix (Page 11).
>
> -
>
> Q) In Table 3, the performance is only measured w.r.t. inference. Given that the most computationally intensive part of the method is the backward pass, a comparison with respect to wall-clock time per epoch, as well as total training time would be very informative.
>
> We agree completely that training time is a key bottleneck. The idea described in this paper, however, cannot immediately provide benefits to reduce training time which is currently comparable to standard attention for the 512 sequence length. However, as the sequence length increases, our method offers both time and memory savings during training. Our main goal was to leverage the effectiveness of the sampling based strategy to minimize the sizable memory footprint that longer sequences entail, which is also a bottleneck. While several groups have approached this in interesting ways, the fact that a backpropagation friendly sampling/hashing based approach can offer significant memory savings, while preserving performance on downstream tasks, is encouraging.
> We have updated the table (Table 4) which is a detailed measurement of time and memory for our method and multiple efficient attention methods with respect to both forward and backward pass. While the runtime efficiency is comparable to other efficient attention methods, the memory savings in all cases is remarkably high which has the potential to enable training on more moderately priced hardware.
>
> -
>
> Q) I find the idea very elegant and interesting however, the experimental section is somewhat lacking. No clear evaluation of the trade-off between speed and performance. The MLM task, although significant and demanding, contains sequences of small length, otherwise why not show a graph of performance vs inference-time.
>
> Given a trained model of our method, we included a plot in the paper (plots in the center of Figure 3) in the paper showing the MLM performance versus the number of hashes used during evaluation. Since the runtime of our method is linear with respect to the number of hashes, the inference time is linearly proportional to the number of hashes. Therefore, the plot of performance versus number of hashes (plots in the center of Figure 3) directly reflect the equivalent relation between performance vs inference time. We have updated the caption to clarify this point. We have now also included plots (Figure 5) of relative error and runtime per token versus sequence length for different numbers of hashes, which should provide additional insight into the behavior of our idea.

---

> ### Author Response · Authors · 2020-11-24
> **Experiments on Long Sequence Tasks**
>
> We thank the reviewers for their comments. Based on the suggestion of Reviewer 3, we finished experiments on WikiHop task that is explicitly designed to evaluate YOSO on longer sequence tasks. Due to limited time/resources, we could try a couple of hyperparameter settings. Nonetheless, the results are promising and consistent with the rest of the paper. This sub-section is now included in blue on page 8 (top). The dev accuracy of WikiHop task is 73.7 for YOSO-128-E (see caption about naming convention in Table 2), which is comparable to 73.8 in Longformer without hyperparameter search but slightly worse than 75.0 that Longformer achieves with hyperparameter search.

---

### Official Review · AnonReviewer2 · 2020-10-28

**Rating:** 6
**Confidence:** 3

**Review:**

This paper tries to improve the efficiency of (multi-head) self-attention by reducing the computational complexity from a quadratic one to a linear one. The authors propose to use Bernoulli sampling to approximate the self-attention's softmax distribution through importance sampling via LSH, which makes a linear cost self-attention possible.

Pros:
1. This paper provides a thorough solution of how to accelerate self-attention via an approximation by Bernoulli sampling and LSH.
2. The experiments show that the proposed approach could achieve considerable speedup while preserving model performance.

Cons:
1. The authors should give a more direct comparison with an important related work the reformer, which also uses LSH to speedup the computation of attention.
2. The experiments were mainly conducted on MLM on GLUE, which lacks generalization among tasks. Adding more tasks such as MT or autoregressive/causal LM would make the experimental part more solid and convincing.



---------
Minors:
- Fig 2, 3 and Tab 1 are not cross-refed in the main body
- Format of citation: seems all of the citations are of this format - authors (year), which is not correct when citations do not act as subjective of objective in the sentence. Please check the format guideline.

---

> ### Author Response · Authors · 2020-11-18
> **Response to Reviewer 2**
>
> We thank the reviewer and answer the main questions below.
>
> -
>
> Q) The authors should give a more direct comparison with an important related work Reformer, which also uses LSH to speedup the computation of attention.
>
> Yes, we have included this comparison in this revision but provide a brief clarification below.
> The basic premise of our experimental setup was that Reformer and other efficient self-attention formulations seek to approximate softmax self-attention, and have not been shown to exceed the performance of softmax self-attention by a significant margin. So, if we can show that a method performs about as well as softmax self-attention on a selection of tasks (MLM, SOP, and GLUE), where we find that the difference is negligible between softmax self-attention and our algorithm, this comparison should be meaningfully applicable to other efficient self-attention algorithms as well.
> Nonetheless, we appreciate this concern and one which can be fully resolved by presenting quantitative results. We have run experiments to compare the performance of Reformer and other efficient self-attention models directly and have updated our paper and report the results on Table 3 in this revision.
>
> -
>
> Q) The experiments were mainly conducted on MLM on GLUE, which lacks generalization among tasks. Adding more tasks such as MT or autoregressive/causal LM would make the experimental part more solid and convincing.
>
> These tasks were chosen primarily because several other efficient self-attention methods have also used these tasks to show viability of their methods. For example, Linformer (Wang et al., 2020) uses MLM pre-training and GLUE tasks (see page 7) to compare their method to softmax self-attention, and Longformer (Beltagy et la., 2020) uses MLM pretraining and multiple NLP tasks (see page 7 and 8) to demonstrate the performance of their method. But we agree that in an ideal setting, if the scope of the experimental setup is much broader, this concern can be fully addressed.
> Regarding extensions to other tasks, we agree that such experiments will be important next steps.  Based on our understanding, autoregressive models are indeed feasible for both Reformer and Longformer but will need adjustments in the implementation. It appears to us that it will be more involved for Linformer. It is, in fact, possible for our algorithm also but will need certain adjustments in the implementation, similar to Reformer and Longformer. Specifically, instead of adding all value vectors into the hashtable, we will need to modify to perform,
> do add v0, query q1, add v1, query q2, add v2, and so on. This will prevent the model looking ahead.
> But even partly studying the viability of our algorithm for these tasks will require, at the very least, a completely fresh set of baselines to convince a reader and will significantly expand the number of experiments and thereby, the resources needed. For these reasons, we believed it was prudent to focus on replicating the experimental setup described in some contemporary papers (such as Linformer (Wang et al., 2020) and Longformer (Beltagy et la., 2020)) that helps highlight the key benefits of our ideas and undertake these interesting next steps in a follow-up work. We hope the reviewer agrees how this would avoid making the resource/budgetary requirements of the current paper untenable.
>
> -
>
> Minors:
> 1.  Fig 2, 3 and Tab 1 are not cross-refed in the main body
> 2. Format of citation: seems all of the citations are of this format - authors (year), which is not correct when citations do not act as subjective of objective in the sentence. Please check the format guideline.
>
> We have fixed these minor issues which are reflected in the updated paper. Thank you.

---

> ### Author Response · Authors · 2020-11-24
> **Experiments on Long Sequence Tasks**
>
> We thank the reviewers for their comments. Based on the suggestion of Reviewer 3, we finished experiments on WikiHop task that is explicitly designed to evaluate YOSO on longer sequence tasks. Due to limited time/resources, we could try a couple of hyperparameter settings. Nonetheless, the results are promising and consistent with the rest of the paper. This sub-section is now included in blue on page 8 (top). The dev accuracy of WikiHop task is 73.7 for YOSO-128-E (see caption about naming convention in Table 2), which is comparable to 73.8 in Longformer without hyperparameter search but slightly worse than 75.0 that Longformer achieves with hyperparameter search.

---

### Official Review · AnonReviewer4 · 2020-10-29
**Attention by LSH sampling**

**Rating:** 5
**Confidence:** 4

**Review:**

This article presents YOSO, an locality sensitive sampling based attention mechanism for large scale language modeling.



Strength:
A new idea of applying  locality sensitive sampling to approximate attention matrix in the transformer


Weakness:
1. Comparison of Complexity: [1] presents the complexity of different efficient transformers. For linformer[2], the time and memory complexity is O(nk). Is there any justification of LSH sampling equipped YOSO with complexity more than O(nm\tau log(d)+nmd)?
2.Experiments: YOSO takes linformer as baselines. However, the pre-training experiment part does not provide steps vs ppl of linformer with YOSO in Figure 4. What is the comparison result of YOSO with linformer on iteration wise convergence? Also, linformer demonstrates better accuracy in downstream tasks such as SST-2. Is there any comparison to an explanation that can analyze this difference in performance?
3.Efficiency: YOSO demonstrates an advantage over linformer and longformer in memory and runtime. However, is there any analysis on why YOSO achieves this superiority with higher complexities? Are there any system-level advantages that YOSO can show?

Some discussions:
Reformer[3] design an attention mechanism that computations are held in the neighbor tokens inside the hash buckets. YOSO also uses hash based sampling to compute attention via neighbor tokens that have high collision probability. On the other hand, linformer introduces a more global view for attention by the low rank projection. Is there any analysis of the local vs global intuition?

[1]Efficient Transformers: A Survey https://arxiv.org/pdf/2009.06732.pdf

[2]Linformer: Self-Attention with Linear Complexity https://arxiv.org/abs/2006.04768

[3] Reformer: The Efficient Transformer https://arxiv.org/abs/2001.04451

---

> ### Author Response · Authors · 2020-11-18
> **Response to Reviewer 4**
>
> We thank the reviewer and answer the main questions below.
>
> -
>
> Q) Complexity: [1] presents complexity of different efficient transformers. For linformer[2], time/memory complexity is $O(nk)$. Any justification of LSH sampling equipped YOSO with complexity more than $O(nm\tau log(d)+nmd)$?
>
> The reviewer will see in [1] (Table 1), the calculation of efficient self-attention complexity treats the feature dimension ($d$) as a constant, so $d$ does not show up in the complexity analysis. In fact, the table shows the complexity of Linformer is $O(n)$ ignoring hyperparameter $k$. We consider complexity of our method as a function of sequence length ($n$), feature dimension ($d$), number of hashes ($m$), so complexity is $O(nm\tau log(d)+nmd)$. If we do not consider $d$ as a variable, then the complexity of our method is $O(nm)$. If we also ignore the hyperparameter of our method, the complexity is $O(n)$. As a result, the complexity of our method is actually lower than many efficient attention methods. For example, $m \in \{16, 32, 64, 128\}$ (our method) is usually smaller than $k \in \{128, 256\}$ (Linformer).
>
> -
>
> Q) Experiments: YOSO takes linformer as baselines. What is the comparison result of YOSO with linformer on iteration wise convergence? Also, linformer demonstrates better accuracy in downstream tasks such as SST-2. Is there any comparison or analysis of this difference in performance?
>
> This is a good point. Linformer follows RoBERTa's pretraining setting, which turns out to have better downstream results than the original BERT pretraining. This is a key reason Linformer also offers better performance on downstream tasks. However, RoBERTa's pretraining is significantly more expensive during training. The reviewer will see that Linformer (see page 6) uses 64 V100 GPUs for pretraining. Such resources are unavailable to most research groups. On more modest hardware resources of 8 V100 (which is also expensive), training time will exceed 80 days (batch size 256 (original BERT) takes around 1s per step, so a batch size of 8K (RoBERTa) takes around 30s per step. Linformer model was trained for 250K steps, which is around 86 days on 8 V100). For this reason, we decided to follow BERT pretraining and were unable to report comparisons to Linformer on # of steps versus perplexity.
> Since this comment is affecting enthusiasm for our work, we trained models with our method and other efficient self-attention methods that use RoBERTa’s pretrained weights as initialization of model (to speed up convergence and cut down the cost), and results of corresponding downstream tasks are updated and shown in Table 3. We see that the results are competitive.
>
> -
>
> Q) Efficiency: YOSO demonstrates an advantage over linformer and longformer in memory and runtime. However, is there any analysis on why YOSO achieves this superiority with higher complexities? System-level advantages that YOSO can show?
>
> As mentioned in Q1, the complexity is actually not higher but lower than many other methods, e.g., Linformer, since the number of hashes is usually less than $k$ in Linformer. But complexity only partly influences runtime. For example, currently GPU memory access time is the bottleneck of our method and potential implementation speedups may be possible which optimize memory latency.
>
> -
>
> Discussions: Any analysis of the local vs global intuition w.r.t. Reformer, Linformer etc?
>
> Yes, Linformer is global in the sense that it compresses $n$ tokens to $k$ summary tokens that serve as a good approximation to self-attention under the conditions described in that paper.
> Reformer and our YOSO can be viewed as “local” but this locality is in terms of dot-products in a high dimensional space. Reformer uses LSH to find near neighbors and then calculates a weighted average among the near neighbors. YOSO directly estimates the weighted average, also derived from LSH principles, but avoids explicitly going through the two step process of Reformer using the ideas on page 5, as noted in the reviews. It is useful to compare this scheme to what softmax self-attention is calculating: it assigns higher attention weights when the dot product between the query and the key is large, which can intuitively be thought of as calculating a weighted average of near-neighbors in a high dimensional space. Therefore, the meaning of “locality” is not much different between self-attention and what Reformer/YOSO implement. Linformer can be interpreted as a different strategy to model this high-dimensional locality.
> We should note a contrast between the semantics of locality discussed above and spatial locality (in terms of the input sequence). Sparse attention methods, such as Longformer and Big bird, use an interesting approach of taking advantage of a known spatial locality in the input sequence (tokens in nearby positions in a sequence are more likely to have a dependency) to prune unnecessary attention entries.

---

> ### Author Response · Authors · 2020-11-24
> **Experiments on Long Sequence Tasks**
>
> We thank the reviewers for their comments. Based on the suggestion of Reviewer 3, we finished experiments on WikiHop task that is explicitly designed to evaluate YOSO on longer sequence tasks. Due to limited time/resources, we could try a couple of hyperparameter settings. Nonetheless, the results are promising and consistent with the rest of the paper. This sub-section is now included in blue on page 8 (top). The dev accuracy of WikiHop task is 73.7 for YOSO-128-E (see caption about naming convention in Table 2), which is comparable to 73.8 in Longformer without hyperparameter search but slightly worse than 75.0 that Longformer achieves with hyperparameter search.

---

### Decision · Program_Chairs · 2021-01-07
**Final Decision**

**Decision:**

Reject

**Comment:**

The paper presents an interesting idea for making self-attention efficient. Several reviewers were not satisfied with the experiments because it did not include runtime and sought after benchmarks.  Rebuttal did a good job of clarifying a few of those with newly added experiments that make the paper stronger. However, the new experiments are in limited settings as well as the real advantage over LSH baselines require more investigation.   This could make a good paper in the future if the experiments are made more rigorous with standard tasks and benchmarks.